# What Is Anthrax?

**DOI:** 10.3390/pathogens11060690

**Published:** 2022-06-16

**Authors:** William A. Bower, Katherine A. Hendricks, Antonio R. Vieira, Rita M. Traxler, Zachary Weiner, Ruth Lynfield, Alex Hoffmaster

**Affiliations:** 1Bacterial Special Pathogens Branch, Centers for Disease Control and Prevention, Atlanta, GA 30329, USA; kah1@cdc.gov (K.A.H.); vht8@cdc.gov (A.R.V.); gna9@cdc.gov (R.M.T.); xxd7@cdc.gov (Z.W.); amh9@cdc.gov (A.H.); 2Minnesota Department of Health, Saint Paul, MN 55155, USA; ruth.lynfield@state.mn.us

**Keywords:** anthrax, *Bacillus anthracis*, *Bacillus tropicus*, *Bacillus cereus*, *Bacillus cereus* biovar anthracis, pathogenesis, plasmids

## Abstract

Anthrax has been feared for its high mortality in animals and humans for centuries. The etiologic agent is considered a potentially devastating bioweapon, and since 1876―when Robert Koch demonstrated that Bacillus anthracis caused anthrax―it has been considered the sole cause of the disease. Anthrax is, however, a toxin-mediated disease. The toxins edema toxin and lethal toxin are formed from protein components encoded for by the pXO1 virulence plasmid present in pathogenic *B. anthracis* strains. However, other members of the *Bacillus cereus* group, to which *B. anthracis* belongs, have recently been shown to harbor the pXO1 plasmid and produce anthrax toxins. Infection with these *Bacillus cereus* group organisms produces a disease clinically similar to anthrax. This suggests that anthrax should be defined by the exotoxins encoded for by the pXO1 plasmid rather than the bacterial species it has historically been associated with, and that the definition of anthrax should be expanded to include disease caused by any member of the *B. cereus* group containing the toxin-producing pXO1 plasmid or anthrax toxin genes specifically.

## 1. Introduction

Anthrax has plagued mankind for eons. The Old Testament descriptions of the fifth (i.e., livestock pestilence) and sixth (i.e., boils on man) Egyptian plagues are consonant with typical symptoms of anthrax [1]. Anthrax was also likely the infamous “Black Bane” that swept through Europe in the 1600s, causing over 60,000 deaths in humans and cattle [2]. More recently, anthrax was responsible for one of the largest zoonotic events in recorded history, causing disease in approximately 10,000 Zimbabweans spanning 1978 through 1980 [3].

Anthrax is also infamous for its potential as a bioweapon. Prior to the Biological Weapons Convention adopted in 1975 that prohibited the research, production, and use of anthrax as a bioweapon, several countries had active anthrax bioweapons programs. After the ban, it is believed that some countries continued producing and stockpiling *B. anthrax* for use as a bioweapon. An accidental release from a Soviet bioweapons facility in 1979 resulting in at least 77 inhalation anthrax cases, of which 66 died, confirmed its lethality as a bioweapon [4]. More recently, in October 2001, letters containing anthrax spores were sent through the United States Postal Service to politicians and media offices in Washington DC, New York, and Florida. The anthrax spores aerosolized from these letters infected at least 22 people, resulting in five deaths [5]. As a result of these recent events and the potential for its future use as a bioweapon, the U.S. government stockpiles antimicrobials, anthrax antitoxin, and vaccines for the prevention and treatment of anthrax following a wide area release of aerosolized *B. anthracis* spores, and provides guidelines for the use of these countermeasures [6]. Anthrax antitoxin is only available from the US Strategic National Stockpile. *Bacillus anthracis*, historically described as the causative agent of anthrax, is a nonmotile, Gram-positive, encapsulated rod first observed in the 1850s by Casimir Davaine and shown to cause anthrax in 1876 by Robert Koch, who elucidated its life cycle [7]. The bacterium has two forms: a vegetative form and a spore form. Because *B. anthracis* can form endospores that are resistant to heat, cold, radiation, desiccation, and disinfectants, it can persist under a variety of environments for prolonged periods of time. It is endemic in Sub-Saharan Africa, the Middle East, Central and Southwest Asia, and Central and South America. Sporadic cases occur in North America and Europe, with most occurring in Eastern and Southern Europe [8]. Naturally occurring human infections with *B. anthracis* are usually secondary to outbreaks in animals and are related to the slaughter and consumption of animals sick or dead from anthrax in countries with food shortages, inadequate veterinary inspection, or low vaccination coverage [9].

Early experiments into anthrax pathogenesis focused on the massive terminal bacteremia, which can reach ~1 × 10^9^ bacterium/mL, as the cause of death [10]. However, bacteria-free serum collected during the terminal phase in guinea pig pathogenesis models and injected into healthy guinea pigs produced death with pathological effects similar to those seen in infection with *B. anthracis* [11]. This suggested that the pathophysiology associated with anthrax was likely toxin-mediated. This hypothesis was confirmed with the discovery that *B. anthracis* harbors two large virulence plasmids, pXO1 and pXO2 [12,13,14]. The pXO1 plasmid encodes for the protein components edema factor (EF), lethal factor (LF), and protective antigen (PA), which form the anthrax toxins. The pXO2 plasmid encodes for the capsule, which helps the bacterium evade the innate host immune response. Virulent stains of *B. anthracis* that have lost the pXO1 plasmid become avirulent and, upon reintroduction of the pXO1 plasmid, become virulent again [15], further demonstrating the toxin-mediated nature of the disease.

During infection, vegetative cells produce EF and PA that combine to form edema toxin and LF and PA that combine to form lethal toxin [16,17]. The PA component binds to cell receptors, which allow the enzymatic components EF and LF to be transported into the cell. Within the cell cytoplasm, LF cleaves and inactivates members of the mitogen-activated protein kinase family and NLRP1; EF rapidly increases cyclic AMP resulting in activation of signaling pathways through protein kinase A. Early in the infection, the toxins target cellular pathways, inhibiting the host’s innate immune responses, such as neutrophil priming, chemotaxis, and chemokine production [16]. By disabling these critical innate immune responses at the site of infection, the bacteria can evade the immune response, disseminate throughout the infected host, and produce massive amounts of toxin [16].

Studies in animal models show that both LF and EF induce vascular shock, but through different mechanisms [18,19]. LF induces a cytokine-independent, nonhemorrhagic vascular collapse resulting in hypoxic necrosis, while EF induces cAMP-mediated vascular dysfunction and hemorrhage. In an animal model that used a slow LF infusion to simulate toxin production during an anthrax infection, the toxin produced progressive hypotension associated with reductions in systemic vascular resistance and left ventricular ejection fraction, findings that may occur in the terminal phase of anthrax in humans [20]. These effects were associated with progressive renal and hepatic dysfunction. Autopsy findings for persons who died from inhalation anthrax are consistent with findings from these studies―with extensive amounts of serosanguinous pleural, abdominal, and pericardial fluid, as well as edema and hemorrhage of the mediastinum and surrounding soft tissues [21,22].

Cutaneous anthrax is the most common form, accounting for >95% of human cases, and results from direct contact with or exposure to infected animals or contaminated animal byproducts [23,24]. It is typically a localized skin infection, often occurring on the face, neck, arms, or hands. The skin lesion starts as a pruritic papule that progresses to a vesicle and then to the classic black necrotic eschar. Uncomplicated cutaneous anthrax has a mortality rate of <2% with treatment [3]. However, the mortality rate can be as high as 30% if localized cutaneous infections progress to systemic anthrax [23]. Ingestion anthrax usually results from consumption of infected meat. There are two forms of ingestion anthrax. The oropharyngeal form is less common and occurs with infection of the oropharynx [25,26]. This can result in neck swelling and respiratory compromise. The gastrointestinal form occurs when the spores germinate in and infect the lower GI tract and can be associated with fever/chills, abdominal pain, nausea/vomiting, ascites, fatigue, diarrhea―which may be bloody―and headache [25,27]. Although the mortality rate for reported gastrointestinal anthrax cases is around 74%, mortality can be substantially mitigated through early treatment [28].

Inhalation anthrax results from the inhalation of aerosolized spores and was historically associated with the processing of wool, hides, or hair from infected animals [29,30]. Currently, inhalation anthrax is a concern because *B. anthracis* spores can be, and indeed have been, used as a bioweapon. Inhalation anthrax may have a biphasic presentation that starts as a mild “viral-like” illness with fever, cough, and fatigue, followed by the sudden onset of severe respiratory distress, dyspnea, and hypoxia 2–3 days later [29,30]. The mortality rate for inhalation anthrax in cases reported since the 1960s is around 72% [31]. Injection anthrax is a relatively new form occurring exclusively in drug users who inject heroin contaminated with *B. anthracis* spores; to date, cases have only been identified in Europe [32]. Symptoms of injection anthrax are similar to those of cutaneous anthrax but are usually associated with deeper tissue infection, resulting in systemic disease. The mortality rate for injection anthrax is around 25% [33]. Meningitis may present as either a primary form of disease or as a secondary complication for 14–37% of cases, depending on the route of transmission. The mortality rate exceeds 90% [31].

Although the clinical presentation of human anthrax is diverse, it has historically been defined as an infection only caused by *B. anthracis*. However, there is mounting evidence that *Bacillus* species other than *B. anthracis* can harbor pXO1 plasmids that produce anthrax toxins. In Ivory Coast in 2001 and Cameroon from 2004 to 2005, there were outbreaks of rapid death in chimpanzees and gorillas, in which *B. anthracis* was initially implicated as the etiologic agent [34,35]. However, molecular analyses showed that the organism exhibited chromosomal characteristics associated with *B. cereus* but contained two virulence plasmids, pXO1 and pXO2, that were almost identical to those in *B. anthracis* [36]. Infection with this variant of *B. cereus*, *Bacillus cereus* biovar *anthracis* (Bcbva), produced a clinical illness similar to that produced by *B. anthracis* in animal models [37]. To date, Bcbva infections have not been documented in humans; however, they have led to fatal infections in other mammals in West and Central Africa, including monkeys, an elephant, duikers, mongooses, and porcupines [38,39]. The known distribution of Bcbva is currently limited to a few countries in West and Central Africa with humid tropical forested environments [40]. However, it is possible that Bcbva infections have gone undetected outside these areas. Most sudden animal deaths in anthrax-endemic areas of sub-Saharan Africa are assumed to be caused by *B. anthracis* based on their presentation with limited diagnostic testing and are not confirmed with laboratory testing that would differentiate Bcbva from *B. anthracis*.

Since 1994, seven persons in the southern United States, all of whom were metal workers, have presented with rapid-onset severe pulmonary infections clinically consistent with systemic anthrax [41]. All had infection with non-*B. anthracis* species within the *B. cereus* group containing the toxin genes typically associated with *B. anthracis*. These *Bacillus* spp., unlike the Bcbva variant, only harbored the toxin-producing pXO1 plasmid. They did not harbor pXO2, which encodes for the poly-d-γ-glutamic acid capsule; however, they did express other plasmid-encoded capsules (e.g., hyaluronic acid and tetrasaccharide). All seven patients were critically ill with pneumonia, and five died [42,43,44,45,46]. In addition to the seven pneumonia cases, there is a case report of a Florida resident with a cutaneous lesion consistent with an anthrax eschar caused by *B. cereus* group bacteria containing anthrax toxin genes [47,48] and a report of a cutaneous infection with the anthrax toxin-producing *B. cereus* strain G9241 in a laboratory worker resulting from a laboratory accident [49]. It is unclear why these naturally occurring cases have only been seen in the Gulf Coast States of the United States, as *B. cereus* group bacteria are ubiquitous in the environment. The seven metalworkers who presented with severe pneumonia were presumedly exposed by inhaling the spore form of the organism. An investigation at the worksite where one of the cases occurred identified bacteria from an environmental sample that matched the clinical isolate from the patient—supporting inhalation from the worksite environment as the route of inoculation [50].

The clinical presentation of these metal workers with pulmonary infections has differed from those of patients with inhalation anthrax, who instead usually present with a mediastinitis. The metal workers consistently had necrotizing and suppurative bronchopneumonias [51]. While acute bronchopneumonia occurs in half of inhalation anthrax cases, it is not necrotizing and is felt to be of hematogenous origin [52], it is unclear why the cases caused by *B. cereus* present differently from those caused by *B. anthracis*. While these *B. cereus* group bacteria produce a capsule, it is possible that the poly-d-γ-glutamic acid capsule encoded by the pXO2 plasmid facilitates evasion of the immune response in a way that favors transport to and germination within the mediastinal lymph nodes rather than the lungs, resulting in the different pathological findings. Since we have only seen these pulmonary cases in metal workers, the different presentation might also be associated with a unique host factor. Indeed, there is evidence that heavy metal exposure can reduce resistance to infection with *B. anthracis* in experimental animals [53], and metal workers have been reported to be more susceptible to other respiratory pathogens [54]. However, the remaining features of these infections—including the eschar skin lesion, the hemodynamic abnormalities, and the high mortality—are almost clinically indistinguishable from disease caused by *B. anthracis* and are likely due to the anthrax toxins produced by these organisms. Furthermore, serum samples taken from a *B. cereus* pneumonia patient who survived and from the Florida cutaneous case showed these patients had detectable levels of lethal factor, anti-protective antigen IgG, and anthrax lethal toxin neutralizing activity [46]. One patient with severe pneumonia received anthrax antitoxin and demonstrated clinical improvement [46]. These findings imply that patients infected with *B. cereus* group bacteria containing anthrax toxin genes should receive anthrax antitoxin in addition to antimicrobials with activity against the *B. cereus* group. Similar to *B. anthracis*, other members of the *B. cereus* group produce beta-lactamases. Thus, empiric therapy should not include penicillin or cephalosporin class antimicrobials. Susceptibility testing can guide treatment options.

## 2. Conclusions

There is recent evidence that *Bacillus* spp. other than *B. anthracis* cause disease mediated by anthrax toxins [41,55]. Thus far, this limited number of cases has been restricted to a specific and potentially more susceptible population (i.e., welders and metal workers); it is not known whether the disease would be of equivalent virulence in persons who lack underlying pulmonary compromise. A limited number of studies in mice suggest that the *B. tropicus* strain G9241 containing the pXO1 plasmid is not as virulent as wildtype *B. anthracis* [42,56]. However, strains such as *B. cereus* biovar *anthracis* that harbor both plasmids are believed to have similar virulence [37]. More research is needed to understand the range of virulence of these organisms, particularly since there are now several lineages within the *B. cereus* group that are known to express anthrax toxin genes.

Anthrax is a diverse set of clinical illnesses (i.e., cutaneous, ingestion, inhalation, injection) historically limited to infections caused by *B. anthracis* harboring both pXO1 and pXO2 [31]. We propose a bigger umbrella―one that covers infections caused by *Bacillus* spp. other than *B. anthracis*―species that have recently been documented to express these toxins and cause infections in humans and animals that are clinically compatible with anthrax. Some recent changes support the inclusion of disease caused by anthrax toxin-producing *Bacillus* spp. as anthrax. In 2017, the US Department of Health and Human Services (DHHS) added Bcbva to the list of select agents and toxins [57]. In 2018, the Council of State and Territorial Epidemiologists added *B. cereus*-expressing anthrax toxins to the national anthrax surveillance case definition [58] because of the clinical and treatment similarities to anthrax caused by *B. anthracis*.

The definition of anthrax should be expanded to include infections with *B. cereus* group organisms expressing anthrax toxins and producing a severe systemic illness characterized by fever or hypothermia, tachycardia, tachypnea, hypotension, leukocytosis, and/or a coagulopathy. Depending on the route of inoculation, clinical features may also include shortness of breath, abdominal pain, nausea/vomiting, headache, or altered mental status. For localized cutaneous anthrax, the definition would be a lesion that progresses from a small, painless, pruritic papule through a vesicular stage into a depressed black eschar, where anthrax toxins or an epidemiological link to anthrax are present. Since the requisite features would be toxin expression and clinically compatible disease, in the future, disease caused by non-*B. cereus* group bacteria might be included.

There are several benefits to using this inclusive definition of anthrax: it would increase awareness of these emerging anthrax toxin-producing *B. cereus* spp. and enhance the detection of such cases. Although anthrax antitoxins are only FDA approved for anthrax, their administration appeared to clinically benefit a welder infected with a *B. cereus* anthrax toxin-producing strain [46]. Expanding the definition would facilitate (1) the use of anthrax antitoxin in conjunction with antimicrobials for treatment in these cases and (2) a risk assessment for using anthrax vaccine for pre- or postexposure prophylaxis in metal workers, i.e., welder anthrax [54].

## Data Availability

No new data were created or analyzed in this study. Data sharing is not applicable to this article.

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
