# Peer review of "What Is Anthrax?"

_pathogens, 2022, doi:10.3390/pathogens11060690_

Round 1

Reviewer 1 Report

It is a very well-written overview of Anthrax and anthrax-causing Bacilli. I hope that the authors will consider the below comments to strengthen their work further.

1. Line 52: what was the host species used in these early experiments? What are the references?

2. Line 68: in addition to cleaving MAPKKs, LF also cleaves host Nlrp1.

3. Line 175: antitoxins as therapy is mentioned for the first time at the end of the manuscript, which may be not well understood by the non-expert readers. Please consider writing a paragraph on FDA-approved anthrax therapies (antitoxins, antibiotics, and a vaccine). 

4. Line 209: "...anthrax antitoxin is only FDA approved for anthrax...": since there are 3 FDA approved antitoxin therapies, perhaps this sentence should be "...anthrax antitoxins are only FDA approved for anthrax...".

5. Something to consider: anthrax antitoxins are approved in combination with antibiotics. If anthrax is caused by non-B. cereus species, the antibiotic administered along with the antitoxins may be different than the one used to treat B. anthracis.

Author Response

1. Line 52: what was the host species used in these early experiments? What are the references?

Answer – Thank you for noting the need for a reference here. Several animals were used in these early experiments. We added the following reference, Zinnser, H. and Bayne-Jones, S. 1939, A Textbook of Bacteriology, D. Appleton-Century Co., Inc., New York.  

2. Line 68: in addition to cleaving MAPKKs, LF also cleaves host Nlrp1.

Answer – Thank you for pointing this out. We added that it cleaves both MAPKKs and NLRP1.

3. Line 175: antitoxins as therapy is mentioned for the first time at the end of the manuscript, which may be not well understood by the non-expert readers. Please consider writing a paragraph on FDA-approved anthrax therapies (antitoxins, antibiotics, and a vaccine). 

Answer – Thank you for noting the lapse in our background information. We have added the following to the end of the second paragraph that discuss B. anthracis as a bioweapon.

“As a result of these recent events and the potential for its future use as a bioweapon, the U.S. government stockpiles antimicrobials, anthrax antitoxin, and vaccine for the prevention and treatment of anthrax following a wide area release of aerosolized B. anthracis spores and provides guidelines for the use of these countermeasures [REF: Bower WA, Yu Y, Person M, Parker C, Kennedy J, Sue D, Hess L, Cook R, Godfred-Cato S, Chatham-Stephens K, Hendricks KA. Prevention and Treatment of Anthrax: CDC Recommendations. MMWR 2022. Submitted]. Anthrax antitoxin is only available from the US Strategic National Stockpile.”

4. Line 209: "...anthrax antitoxin is only FDA approved for anthrax...": since there are 3 FDA approved antitoxin therapies, perhaps this sentence should be "...anthrax antitoxins are only FDA approved for anthrax...".

Answer – Thank you. We changed to “antitoxins are.”

5. Something to consider: anthrax antitoxins are approved in combination with antibiotics. If anthrax is caused by non-B. cereus species, the antibiotic administered along with the antitoxins may be different than the one used to treat B. anthracis.

Answer – Thank you for raising this important point. We considered this as well and provided the following, which we believe covers the issue, “These findings imply that patients with infections with B. cereus group bacteria containing anthrax toxin genes should receive anthrax antitoxin in addition to antimicrobials with activity against the B. cereus group. Similar to B. anthracis, other members of the B. cereus group produce beta-lactamases, thus empiric therapy should not include penicillin or cephalosporin class antimicrobials. Susceptibility testing can guide treatment options.”

Reviewer 2 Report

A succinct summary of the anthrax disease definition and presentation issue. It is a good piece to recommend an update of definitions for other policy makers and disease control programs.

Author Response

A succinct summary of the anthrax disease definition and presentation issue. It is a good piece to recommend an update of definitions for other policy makers and disease control programs.

Answer: Thank you for taking the time to review this manuscript. There does not appear to be a request for edits.

Reviewer 3 Report

This work by Bower and colleagues attempts to redefine and expand the anthrax. The central claim of this study is to expand the definition of anthrax based on plasmid PXO1 harbored by Bacillus and the spectrum of illness. In fact, this perspective has been discussed and reviewed in detail elsewhere (Baldwin, 2020, Frontiers in Microbiology, PMID: 32973690). With the high similarities to the Baldwin review article, novel perspectives and up-to-date new research studies needs to be incorporated to distinguish this article from the existing one.

1.     Reference the Baldwin paper as there are so much similarities between them.

2.     As Baldwin paper and other studies suggests, both toxin-encoded plasmid (pXO1) and capsule are required for toxicity and/or anthrax-like symptoms. It is appropriate to take into consideration both features to redefine the anthrax. In addition, please summarize the Bacillus spp that cause anthrax-like symptoms based on their harboring plasmid and capsule (poly-D-r-glutamic acid capsule, hyaluronic acid and tetrasaccharide).  Because Baldwin paper comprehensively summarized the Bacillus spp. that causes anthrax-like symptoms with correlation to both pXO1 and capsule features, the authors needs to incorporate new strains if there is any and present them in a way that is different than the Baldwin table. 

3.     Please describe or review in detail about the significance of redefining the anthrax as this is the major point of this review/perspective. However, the amount of discussion is much less than introducing the history of anthrax.

4.     To redefine anthrax, the authors need to put forward a relatively clear and detailed standard/criteria of doing so.

5.     As a perspective/review article, there are lots of references that need to be incorporated to support the statement/description and appreciate the contributions of the previous study. Please add appropriate references to the sentences in line 50, line 52, line 58, line 60, line 61, line 66, line 68, line 72, line 76, line 81, line 89, line 93, line 95, line 99, line 103, line 107, line 108, line 110, line 112, line 113, line 126, line 137, line 163, line 174, line 184, line 187, line 190, line 192, line 195 and line 205.

Author Response

  1. Reference the Baldwin paper as there are so much similarities between them.

Answer: Thank you for asking about the Baldwin paper. This paper very much supports a shift in the anthrax definition. We didn’t original use it because we provide the original references describing these cases rather than the summary as in the Baldwin paper. Also, the table in the Baldwin paper states the G9898 isolate as having pBCXO1. Neither of the references, Miller et al and Sue et al, describe this isolate as having pBCXO1. However, we did find a place to include it in the conclusion section.

  1. As Baldwin paper and other studies suggests, both toxin-encoded plasmid (pXO1) and capsule are required for toxicity and/or anthrax-like symptoms. It is appropriate to take into consideration both features to redefine the anthrax. In addition, please summarize the Bacillus spp that cause anthrax-like symptoms based on their harboring plasmid and capsule (poly-D-r-glutamic acid capsule, hyaluronic acid and tetrasaccharide).  Because Baldwin paper comprehensively summarized the Bacillus spp. that causes anthrax-like symptoms with correlation to both pXO1 and capsule features, the authors need to incorporate new strains if there is any and present them in a way that is different than the Baldwin table.

Answer: Thank you for your input on the definition of anthrax. We disagree with this position. We believe that anthrax is a toxin-mediated disease. While capsules can help the organism evade the host immune system and are an important virulence factor for B. anthracis and many other pathogens, it does not define anthrax. This is supported by studies that show an anthrax-like disease can be induce in animal models by infusion of anthrax toxin alone.  

  1. Please describe or review in detail about the significance of redefining the anthrax as this is the major point of this review/perspective. However, the amount of discussion is much less than introducing the history of anthrax.

Answer: Thank you for your thorough review of our manuscript. We believe our conclusion is succinct and fully describes our perspective. We also list several benefits to a change in the definition. See below.

“There are several benefits to using this inclusive definition for anthrax: It would increase awareness of these emerging anthrax toxin-producing B. cereus spp. and enhance detection of such cases. Although anthrax antitoxins are only FDA approved for anthrax, their administration appeared to clinically benefit a welder infected with a B. cereus anthrax toxin-producing strain. Expanding the definition would facilitate 1) the use of anthrax antitoxin in conjunction with antimicrobials for treatment in these cases and 2) a risk assessment for using anthrax vaccine for pre- or postexposure prophylaxis in metal workers, i.e., welder anthrax.”

  1. To redefine anthrax, the authors need to put forward a relatively clear and detailed standard/criteria of doing so.

Answer: Thank you for pointing out this omission. We added the following case definition for anthrax that includes the presence of anthrax toxins and clinical features that taken from the CDC surveillance case definition.

“The definition for anthrax should be expanded to include infections with B. cereus group organisms expressing anthrax toxins and producing a severe systemic illness characterized by fever or hypothermia, tachycardia, tachypnea, hypotension, leukocytosis, and/or a coagulopathy. Depending on the route of inoculation clinical feature may also include shortness of breath, abdominal pain, nausea/vomiting, headache or altered mental status. For localized cutaneous anthrax, the definition would be a lesion that progresses from a small, painless, pruritic papule, through a vesicular stage into a depressed black eschar and has presence of anthrax toxin or epidemiological link to anthrax.”

  1. As a perspective/review article, there are lots of references that need to be incorporated to support the statement/description and appreciate the contributions of the previous study. Please add appropriate references.

Answer: We added the following references.

line 50 - Anthrax in Humans and Animals (World Health Organization and International Office of Epizootics, 2008)

line 52 - Zinnser, H. and Bayne-Jones, S. 1939, A Textbook of Bacteriology, D. Appleton-Century Co., Inc., New York

line 58 – Uchida I, Hashimoto K, Terakado N. Virulence and immunogenicity in experimental animals of Bacillus anthracis strains harbouring or lacking 110 MDa and 60 MDa plasmids. J Gen Microbiol. 1986 Feb;132(2):557-9; Green BD, Battisti L, Koehler TM, Thorne CB, Ivins BE. Demonstration of a capsule plasmid in Bacillus anthracis. Infect Immun. 1985 Aug;49(2):291-7; Keppie J, Harris-Smith PW, Smith H. The Chemical Basis of the Virulence of Bacillus Anthracis. IX. Its Aggressions and Their Mode of Action. Br J Exp Pathol. 1963 Aug;44(4):446-53

line 60 and 61 – Referenced on line 58.

line 66 and line 68 – Moayeri, M.; Leppla, S.H.; Vrentas, C.; Pomerantsev, A.P.; Liu, S. Anthrax Pathogenesis. Annu Rev Microbiol 2015, 69, 185-208, doi:10.1146/annurev-micro-091014-104523; Liu, S.; Moayeri, M.;, Leppla, S.H. SH, Vrentas C, Pomerantsev AP, Liu S. Anthrax lethal and edema toxins in anthrax pathogenesis. Trends Microbiol 2014, 22, 317-325, doi:10.1016/j.tim.2014.02.012.pathogenesis. Annu Rev Microbiol. 2015 Jul 16;69(1):185-208.

line 72 – Liu, S.; Moayeri, M.; Leppla, S.H. Anthrax lethal and edema toxins in anthrax pathogenesis. Trends Microbiol 2014, 22, 317-325, doi:10.1016/j.tim.2014.02.012.

line 76 – Moayeri, M.; Haines, D.; Young, H.A.; Leppla, S.H. Bacillus anthracis lethal toxin induces TNF-alpha-independent hypoxia-mediated toxicity in mice. J Clin Invest 2003, 112, 670-682, doi:10.1172/JCI17991; Firoved, A.M.; Miller, G.F.; Moayeri, M.; Kakkar, R.; Shen, Y.; Wiggins, J.F.; McNally, E.M.; Tang, W.J.; Leppla, S.H. Bacillus anthracis edema toxin causes extensive tissue lesions and rapid lethality in mice. Am J Pathol 2005, 167, 1309-1320, doi:10.1016/S0002-9440(10)61218-7.

line 81 – Sweeney, D.A.; Cui, X.; Solomon, S.B.; Vitberg, D.A.; Migone, T.S.; Scher, D.; Danner, R.L.; Natanson, C.; Subramanian, G.M.; Eichacker, P.Q. Anthrax lethal and edema toxins produce different patterns of cardiovascular and renal dysfunction and synergistically decrease survival in canines. J Infect Dis 2010, 202, 1885-1896, doi:10.1086/657408

line 89 – Fasanella A, Galante D, Garofolo G, Jones MH. Anthrax undervalued zoonosis. Vet Microbiol. 2010;140:318–31. 10.1016/j.vetmic.2009.08.016; Doganay M, Metan G, Alp E. A review of cutaneous anthrax and its outcome. J Infect Public Health. 2010;3:98–105. 10.1016/j.jiph.2010.07.004

line 93 – Doganay M, Metan G, Alp E. A review of cutaneous anthrax and its outcome. J Infect Public Health. 2010;3:98–105. 10.1016/j.jiph.2010.07.004

line 95 – Sirisanthana T, Navacharoen N, Tharavichitkul P, et al. Outbreak of oral-pharyngeal anthrax: an unusual manifestation of human infection with Bacillus anthracis. Am J Trop Med Hyg. 1984;33:144-150.; Beatty ME, Ashford DA, Griffin PM, Tauxe RV, Sobel J. Gastrointestinal anthrax: review of the literature. Archives of Internal Medicine. 2003 Nov 10;163(20):2527-31.

line 99 – Amidi S, Dutz W, Kohout E, Ronaghy A. Human anthrax in Iran: report of 300 cases and review of literature. Z Tropenmed Parasitol. 1974;25:96-104; Beatty ME, Ashford DA, Griffin PM, Tauxe RV, Sobel J. Gastrointestinal anthrax: review of the literature. Arch Intern Med. 2003;163(20):2527-2531.

line 103 – Brachman PS. Inhalation anthrax. Ann N Y Acad Sci 1980; 353:83-93; Laforce FM. Woolsorters' disease in England. Bulletin of the New York Academy of Medicine. 1978 Nov;54(10):956.

line 107 – Brachman PS. Inhalation anthrax. Ann N Y Acad Sci 1980; 353:83-93; Laforce FM. Woolsorters' disease in England. Bulletin of the New York Academy of Medicine. 1978 Nov;54(10):956.

line 108 – Hendricks K, Person MK, Bradley JS, et al. Anthrax: a comprehensive review describing the clinical features of reported hospitalized cases for all routes of infection published in the English literature, 1880-2018. Clin Infect Dis 2022; Submitted

line 110 – Booth MG, Hood J, Brooks TJ, Hart A; Health Protection Scotland Anthrax Clinical Network. Anthrax infection in drug users. Lancet. 2010;375(9723):1345-1346

line 112 – Booth MG, Hood J, Brooks TJ, Hart A; Health Protection Scotland Anthrax Clinical Network. Anthrax infection in drug users. Lancet. 2010;375(9723):1345-1346

line 113 – National Anthrax Outbreak Control Team. An Outbreak of Anthrax Among Drug Users in Scotland, December 2009 to December 2010. Health Protection Scotland; 2011 [6/12/2022]; Available from: https://www.hps.scot.nhs.uk/web-resources-container/an-outbreak-of-anthrax-among-drug-users-in-scotland-december-2009-to-december-2010-a-report-on-behalf-of-the-national-anthrax-outbreak-control-team/.

line 126 – Brezillon, C.; Haustant, M.; Dupke, S.; Corre, J.P.; Lander, A.; Franz, T.; Monot, M.; Couture-Tosi, E.; Jouvion, G.; Leendertz, F.H.; et al. Capsules, toxins and AtxA as virulence factors of emerging Bacillus cereus biovar anthracis. PLoS Negl Trop Dis 2015, 9, e0003455, doi:10.1371/journal.pntd.0003455

line 137 – Dawson P, Schrodt CA, Feldmann K, et al. Notes from the Field: Fatal Anthrax Pneumonia in Welders and Other Metalworkers Caused by Bacillus cereus Group Bacteria Containing Anthrax Toxin Genes - U.S. Gulf Coast States, 1994-2020. MMWR Morb Mortal Wkly Rep. 2021;70(41):1453-1454. Published 2021 Oct 15. doi:10.15585/mmwr.mm7041a4

line 163 – This doesn’t need a reference. This is speculation on the part of the authors and is prefaced by, “it is possible that.”

line 174 – Kate Hendricks KA, Martines R, Bielamowicz H, Boyer AE, Long S, Byers P, Stoddard RA,  Taylor K, Beesley-Kolton C, Gallegos-Candela M, Roberts C, DeLeon-Carnes M, Salzer J, Dawson P, Brown D, Maves RC, Gulvik C, Lonsway D, Barr, JR, Bower WA, Hoffmaster A. Welder’s Anthrax: A Tale of Two Cases. Clin Inf Dis. 2022 Submitted. 

line 184 – Dawson P, Schrodt CA, Feldmann K, et al. Notes from the Field: Fatal Anthrax Pneumonia in Welders and Other Metalworkers Caused by Bacillus cereus Group Bacteria Containing Anthrax Toxin Genes - U.S. Gulf Coast States, 1994-2020. MMWR Morb Mortal Wkly Rep. 2021;70(41):1453-1454. Published 2021 Oct 15. doi:10.15585/mmwr.mm7041a4; Baldwin VM. You Can't B. cereus - A Review of Bacillus cereus Strains That Cause Anthrax-Like Disease. Front Microbiol. 2020;11:1731. Published 2020 Aug 19. doi:10.3389/fmicb.2020.01731.

line 187 – Doesn’t need a ref here. This is author’s opinion.

line 190 – Brézillon, C., Haustant, M., Dupke, S., Corre, J. P., Lander, A., Franz, T., Monot, M., Couture-Tosi, E., Jouvion, G., Leendertz, F. H., Grunow, R., Mock, M. E., Klee, S. R., & Goossens, P. L. (2015). Capsules, toxins and AtxA as virulence factors of emerging Bacillus cereus biovar anthracis. PLoS neglected tropical diseases, 9(4), e0003455. https://doi.org/10.1371/journal.pntd.0003455

line 192 – References above for line 184

line 195 – Hendricks KA, Person MK, Bradley JS, Mongkolrattanothai T, Hupert N,  Eichacker P, Friedlander A, Bower WA. A Comprehensive Review Describing the Clinical Features of Reported Hospitalized Cases for All Routes of Infection Published in the English Literature, 1880-2018. Clin Inf Dis. 2022. Submitted.

line 205 – Doesn’t need a reference. This is author’s opinion.

Round 2

Reviewer 3 Report

The authors put decent effort on addressing reviewer's comments and the revised version is much more improved. The topic and significance of this review article will be appreciated by the anthrax field including both basic research scientists, policy maker as well as clinical physicians.